# BRAIN ENCODING MODELS BASED ON BINDING MULTIPLE MODALITIES ACROSS AUDIO, LANGUAGE, AND VISION

## ABSTRACT

Multimodal associative learning of sensory stimuli (images, text, audio) has created powerful representations for these modalities that work across a multitude of tasks with simple task heads without even (fine)tuning features on target datasets. Such representations are being increasingly used to study neural activity and understand how our brain responds to such stimuli. While previous work has focused on static images, deep understanding of a video involves not just recognizing the individual objects present in each frame, but also requires a detailed semantic description of their interactions over time and their narrative roles. In this paper, we seek to evaluate whether new multimodally aligned features (like ImageBind) are better than previous ones in explaining fMRI responses to external stimuli, thereby allowing for a better understanding of how the brain and its different areas process external stimuli, converting them into meaningful high-level understanding, and actionable signals. In addition, we explore whether generative AI based modality conversion helps to disentangle the semantic part of the visual stimulus allowing for a more granular localization of such processing in the brain. Towards this end, given a dataset of fMRI responses from subjects watching short video clips, we first generate detailed multi-event video captions. Next, we synthesize audio from these generated text captions using a text-to-speech model. Further, we use a joint embedding across different modalities (audio, text and video) using the recently proposed ImageBind model. We use this joint embedding to train encoding models that predict fMRI brain responses. We infer from our experimental findings and computational results that the visual system's primary goal may revolve around converting visual input into comprehensive semantic scene descriptions. Further, multimodal feature alignment helps obtain richer representations for all modalities (audio, text and video) leading to improved performance compared to unimodal representations across well-known multimodal processing brain regions.

## 1 INTRODUCTION

The increasing availability of naturalistic fMRI datasets and the use of large-scale neural models enable a better understanding of the brain's response to natural stimuli. This line of work, namely brain encoding, aims at predicting the neural brain activity recordings given an input stimulus. While current brain encoding studies are usually trained and tested on brain responses to a single stimulus modality, such as language (Wehbe et al., 2014; Jain & Huth, 2018; Toneva & Wehbe, 2019; Caucheteux & King, 2020; Schrimpf et al., 2021; Oota et al., 2022c;a; Toneva et al., 2022; Aw & Toneva, 2022), vision (Schrimpf et al., 2018) or speech (Millet et al., 2022; Vaidya et al., 2022; Tuckute et al., 2022; Oota et al., 2023a;b), the human brain is remarkable in its ability to integrate information across multiple modalities. There is growing evidence that the ability for multimodal processing is underpinned by synchronized cortical representations of identical concepts across various sensory modalities. For instance, it has been shown that the semantic scene textual descriptions could offer a more effective characterization of visually induced activity (Doerig et al., 2022; Oota et al., 2022b; Popham et al., 2021; Wang et al., 2022). However, these studies have experimented with subjects watching static images only.

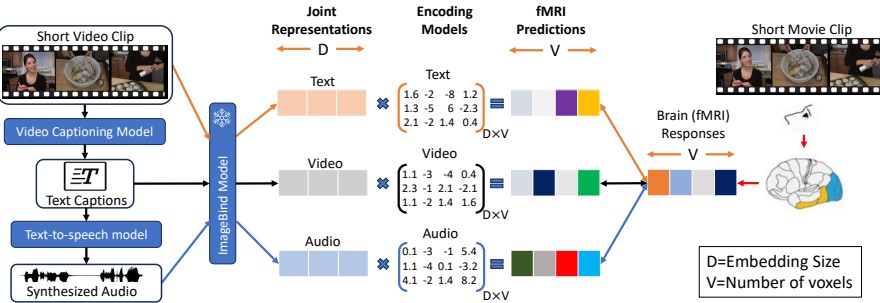

Figure 1: Overview of our proposed Multimodal Brain Encoding Pipeline

Deep understanding of a video requires temporal semantics, involving object kinematics, their interactions and dynamics in the broader context of visual narratives. To obtain a better understanding of how different areas of the brain process video stimuli, we first seek to evaluate whether new multimodally (video, audio, text) aligned features are powerful enough to be used to extract meaningful information from the fMRI experiments with humans subjected to the video stimuli. Hence, in this work, we extend the multimodal brain encoding line of work along two aspects: (1) rather than static image stimuli, we experiment with richer video stimuli with multiple temporal events, and (2) rather than text-only semantic scene descriptions (at image level), we leverage text descriptions (at video level). In this work, we perform alignment between multimodal (joint embedding space) representations and fMRI brain responses by training brain encoding models to verify whether these richer semantic scene descriptions provide better characterization of visually evoked activity.Encoding models that successfully predict fMRI brain activity across different regions of the brain can provide insights into how the human brain actively engages in visuo-semantic transformations while watching short clips.

Clearly, embedding videos is more challenging than embedding images. Also, leveraging text+audio descriptions along with videos for brain encoding needs a novel architecture. One way to extract aligned features of audio, language and visual stimuli is using transformer models trained on multimodal objectives like audio-image-text matching. Recent studies have shown that multimodal transformers outperform unimodal transformers at modeling brain responses to language and visual stimuli, suggesting that multimodal training enables models to learn more brain-like representations (Oota et al., 2022b; Tang et al., 2023; Dong & Toneva, 2023). However, these studies assess representations from multimodal transformers for static images and their associated captions only.

Recently generative AI advancements have produced high-performing effective models for modality conversion as well as for multimodal representation learning. Popular models for video to text generation include MiniGPT4 (Zhu et al., 2023), mPlugOwl (Ye et al., 2023) and Video-LLaMA (Zhang et al., 2023). Similarly, audio synthesis can be done effectively using MMS () or SeamlessM4T models (Barrault et al., 2023). Lastly, multimodal representations can be learned using VisualBERT (Li et al., 2019), vilBERT (Lu et al., 2019), CLIP (Radford et al., 2021), ImageBind Girdhar et al. (2023), etc. Can generative AI based modality conversion help obtain richer video stimuli representations enabling better understanding of the brain multimodal processing pathway? As shown in Fig. 1, given a dataset of fMRI responses from subjects watching short video clips, we first generate detailed multi-event video captions and then synthesize audio from these generated text captions using a text-to-speech model. Further, we leverage the pretrained ImageBind (Girdhar et al., 2023) model to jointly learn embeddings across text and video.

Are such multimodal-bound representations better aligned with fMRI brain responses? To answer this question, we estimated encoding models using ImageBind features of the three modalities (text, audio, video) separately, and predict fMRI responses for the corresponding short clips. We evaluate how well each of the three encoding models predict fMRI responses while subjects were engaged in passively watching short clips. This analysis also helps us identify brain regions responsible for processing multimodal information.

Overall, we make the following contributions in this paper. (1) To the best of our knowledge, this is the first work on leveraging unified multimodal representations of video stimuli for brain encoding.

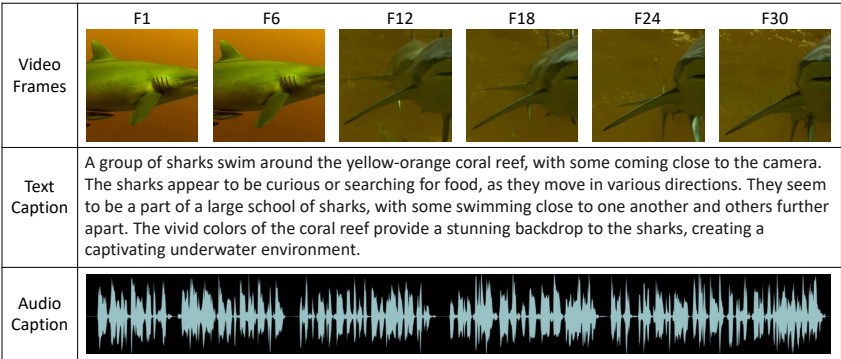

Table 1: Generated text caption and synthesized audio for a sample short video clip.

(2) We evaluate the performance of these joint multimodal representations in predicting brain activity at the level of whole brain as well as we investigate this at a more granular level, wherein we perform region-wise analysis. For each video clip, we harness popular generative AI models to generate text video captions and use them to synthesize audio. ImageBind helps us to multimodally align video captions and synthesized audio with video clips.

(3) Specifically, we attempt to answer which brain regions process what kind of visual information, by disentangling the semantic information from the videos.

(4) Experiments show that joint multimodal embeddings learned using ImageBind lead to improved predictivity of brain activations especially in well-known multimodal areas like dorsal visual (action pathways), middle temporal (visual attention) and late language.

## 2 DATASET CURATION

**Brain Imaging Dataset** In this paper, we experiment with the "Short Movie Clips" fMRI dataset (Huth et al., 2022) which was collected while human subjects were passively watching short movie clips. This is one of the largest publicly available fMRI dataset in terms of number of samples per participant. The dataset contains data from five subjects who watched 10 English short movie clips, and each movie clip is about 10 minutes long. For each subject, the dataset contains 3600 samples in train and 271 samples in test. The fMRI data is collected every 2 seconds ( Time Repition - TR ) spanning over 30 frames.

The dataset is already preprocessed and projected onto the surface space ("fsaverage"). We use the multimodal parcellation of the human cerebral cortex (Glasser Atlas consisting of 180 ROIs in each hemisphere) to report the ROI analysis and PyCortex for the brain maps (Glasser et al., 2016). The data covers seven brain regions of interest (ROIs) in the human brain with the following subdivisions: (i) primary visual (PV: V1); (ii) early visual (EV: V2, V3 and V4); (iii) dorsal visual (DV: V3A, V3B, V6, V6A, V7 and IPS1); (iv) ventral visual (VV: V8, VMV1, VMV2, VMV3, VVC and FFC); (v) visual word form area (VWFA: PH and TE2P); (vi) middle temporal (MT: MT, MST, LO1, LO2, LO3, FST and V3CD), (vii) late language (LL: A5, 44, 45, IFJa, IFSp, PGi, PGp, PGs, TPOJ1, TPOJ2, TPOJ3, STGa, STS, TA2) (Baker et al., 2018; Milton et al., 2021; Desai et al., 2022). We report the functionality of these brain regions in the Appendix Section C.

**Video Captioning and Audio synthesis** To generate text captions for each 2 second clip, we use the mPlugOwl model (Ye et al., 2023). To generate audio from the caption, we use the text-to-speech (TTS) from the Massively Multilingual Speech (MMS) model (Pratap et al., 2023), open-sourced by Meta. Table 1 shows generated text caption and synthesized audio for the corresponding video clip.

**Multimodal Embeddings using ImageBind** Recently, a multimodal model called Image-Bind (Girdhar et al., 2023) showed immense promise in binding data from six modalities at once, without the need for explicit supervision. We extract common embedding representations for the brain encoding task. We input video, synthesized audio and the generated text caption at each TR (repetition time) and obtain the aligned embeddings for the three modalities. We use the pre-trained ImageBind Transformer model which outputs a 1024 dimensional representation for each modality.

**Embeddings for Individual Modalities** To investigate the effectiveness of multimodal representations in comparison to representations for individual modalities, we use the following methods to obtain embeddings for individual modalities. We use the BERT-base-uncased (Devlin et al., 2019) model[1] as our text-only encoder model for extracting representations for text captions. To extract text embeddings, we input the generated text captions to the pretrained BERT model which outputs 768 dimensional vector for each caption. AST (Audio Spectrogram Transformer) (Baade et al., 2022), the underlying ImageBind audio encoder model is used for extracting representations for audio captions. To extract audio embeddings, we input the audio wav file (obtained from the TTS model of MMS) to AST model. It outputs 768 dimensional vector for each caption. ViT-H (Vision Transformer Huge) (Dosovitskiy et al., 2020), the underlying video encoder model for ImageBind is used for extracting representations for frames in each video. To extract embedding at each TR, we average all frame embeddings and obtain the corresponding video representation.

**Downsampling** Since the rate of fMRI data acquisition (TR = 2 seconds) in the dataset is lower than the rate at which the stimulus was presented to the subjects (15 frames per second), 30 frames of a video were viewed under the same TR for a single fMRI acquisition. This helps us to attain synchronization between the stimulus presentation rate and fMRI data recording, which we then leverage to train our encoding models.

**TR Alignment** For each subject, we account for the delay in the hemodynamic response by modeling hemodynamic response function using a finite response filter (FIR) per voxel with 5 temporal delays (TRs) corresponding to 10 seconds (Huth et al., 2022).

## 3 METHODOLOGY

**Encoding Model** We train banded ridge regression based voxel-wise encoding models Deniz et al. (2019) to predict the fMRI brain activity associated with the stimulus representations obtained from the individual modalities (text, audio and video) and joint embeddings from ImageBind model.

Formally, at each time step $t$, we encode the stimuli as $X_t \in \mathbb{R}^D$ and brain region voxels $Y_t \in \mathbb{R}^V$, where $D$ denotes the dimension of the concatenation of delayed 5 TRs, and $V$ denotes the number of voxels. Overall, with $N$ such TRs, we obtain $N$ training examples.

**Train-test Setup** All the data samples from 11 training sessions (3600 TRs) were used for training, and the generalization was tested on samples from the test sessions (271 TRs).

**Evaluation Metrics** We evaluate our models using Pearson Correlation (PC) which is a standard metric for evaluating brain alignment (Jain & Huth, 2018; Schrimpf et al., 2021; Goldstein et al., 2022). Let TR be the number of time repetitions in the test set. Let $Y = \{Y_i\}_{i=1}^{TR}$ and $\hat{Y} = \{\hat{Y}_i\}_{i=1}^{TR}$ denote the actual and predicted value vectors for a single voxel. Thus, $Y \ and \ \hat{Y} \in \mathbb{R}^{TR}$. We use Pearson Correlation (PC) which is computed as $corr(Y, \hat{Y})$ where corr is the correlation function.

**Noise Ceiling and Explained Variance** To account for the intrinsic noise in biological measurements and obtain a more accurate estimate of the model's performance, we estimate the explained variance from repeated stimuli of test dataset, as reported in (Huth et al., 2022). The neural model predictivity voxels were considered by selecting significant voxels whose explained variance is > 0.1. The final measure of a model's performance on a dataset is thus the average Pearson correlation of significant voxels i.e., the model variance explained by individual subject's significant voxels.

**Implementation Details for Reproducibility** All experiments were conducted on a machine with 1 NVIDIA GeForce-GTX GPU with 16GB GPU RAM. We used banded ridge-regression with the following parameters: MSE loss function; L2-decay ($\lambda$) varied from $10^{-1}$ to $10^{-3}$; the best $\lambda$ was chosen by tuning on validation data that comprised a randomly chosen 10% subset from the train set used only for hyper-parameter tuning. For BERT, we use layer-7 outputs for extracting the caption representations. We experimented with outputs from each layer, and found that layer 7 provides the best results on the validation set. While training encoding models, the pretrained ImageBind is not finetuned.

---

[1]We did not use the underlying text encoder for ImageBind (CLIP-text) since ImageBind freezes the text encoder.

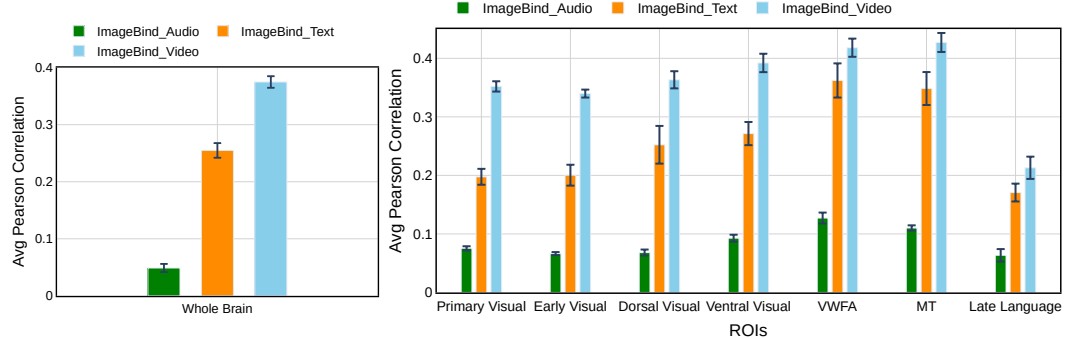

Figure 2: Brain alignment of separately estimated audio, language and vision encoding models averaged across all the subjects using ImageBind representations. The *left plot* compares the average Pearson correlation across all subjects and all voxels (i.e. whole brain). Error bars indicate the standard error of the mean across participants. The dotted horizontal line reports the average explained variance across subjects and voxels. The *right plot* compares the region-wise performance of the three encoding models.

## 4 RESULTS

### 4.1 PERFORMANCE OF MULTIMODALITY EMBEDDINGS

We estimate modality-specific encoding models for each subject using ImageBind embeddings. The audio encoding model uses the ImageBind audio features to predict the fMRI response per voxel for the corresponding clip. Similarly, for the text and vision encoding models, we use the ImageBind text and vision features respectively to predict the fMRI response per voxel for the corresponding clip. For all three modalities (audio, language, and vision), we report the Pearson correlation averaged across all the subjects at whole brain (left) and region-level (right) in Fig. 2.

We make the following observations at whole brain-level from Fig. 2 (left). Compared to audio and text models, ImageBind video exhibits a higher level of brain predictability, in line with our expectations. Since the subjects were involved in watching the movie clips, so the video representations are expected to align best with the fMRI brain activity. Interestingly, we conclude that ImageBind text representations effectively capture rich semantic scene descriptions even when subjects are actively engaged in passive viewing. This implies that the model is able to decouple the semantic information from the video / visual stimuli effectively. Furthermore, we note a positive Pearson correlation in the case of audio representations in the MT region ( visual attention ), which corroborates the findings in audio-visual cognition. Scheef et al. (2009)

From the region-level analysis depicted in Fig. 2 (right), we observe a positive Pearson correlation across all regions of interest (ROIs) for all the three modalities. Across all regions, video representations perform better compared to text or audio. Notably, in early visual, dorsal, and ventral visual areas, there is a more pronounced difference in prediction performance between text and video representations. However, for ROIs such as VWFA, MT, and late language regions, both modalities exhibit high brain predictivity. We also observe that ImageBind text representations are significantly better than explained variance across high-level visual (dorsal and ventral) and language ROIs. For higher-level visual areas, ImageBind audio representations do not work very well but both ImageBind text as well as ImageBind video have high performance. This suggests that those regions may be implicated in processing higher-level visual semantics, especially, that is represented well linguistically.

Moreover, our results are inline with previous studies that regions such as Dorsal Visual, VWFA and MT regions are likely responsible for processing multimodal information (Rolls et al., 2023).

For detailed analysis, for subject 1, we project the Pearson correlation values obtained using the three encoding models (text, audio and video) onto a flattened cortical surface, as shown in Fig. 3. We also illustrate such flatmaps for other subjects in the appendix.

Çelik et al. (2021) have shown that dorsal regions like posterior superior temporal sulcus (pSTS), fusiform face area (FFA), ventral region extrastriate body area (EBA) and human MT (V5/MT+) are

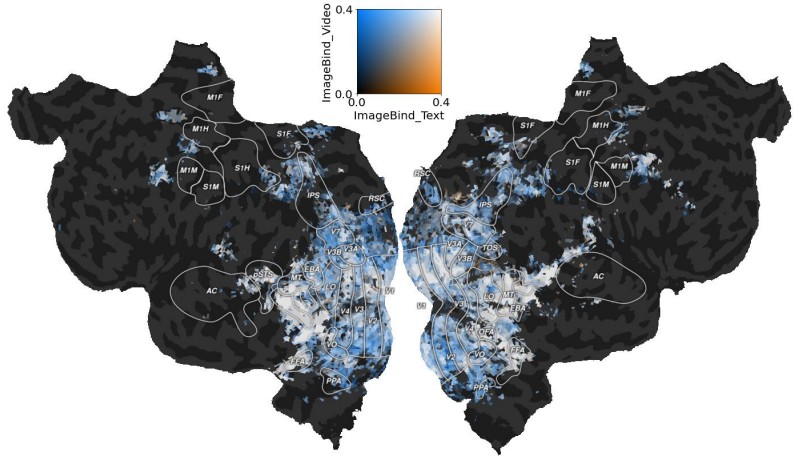

(a) Contrast between ImageBind text performance and video performance for each voxel in one subject (subject-1) is projected onto the subject's flattened cortical surface. Voxels appear orange if they are better predicted by text, blue if they are better predicted by video, and white if they are well predicted by both.

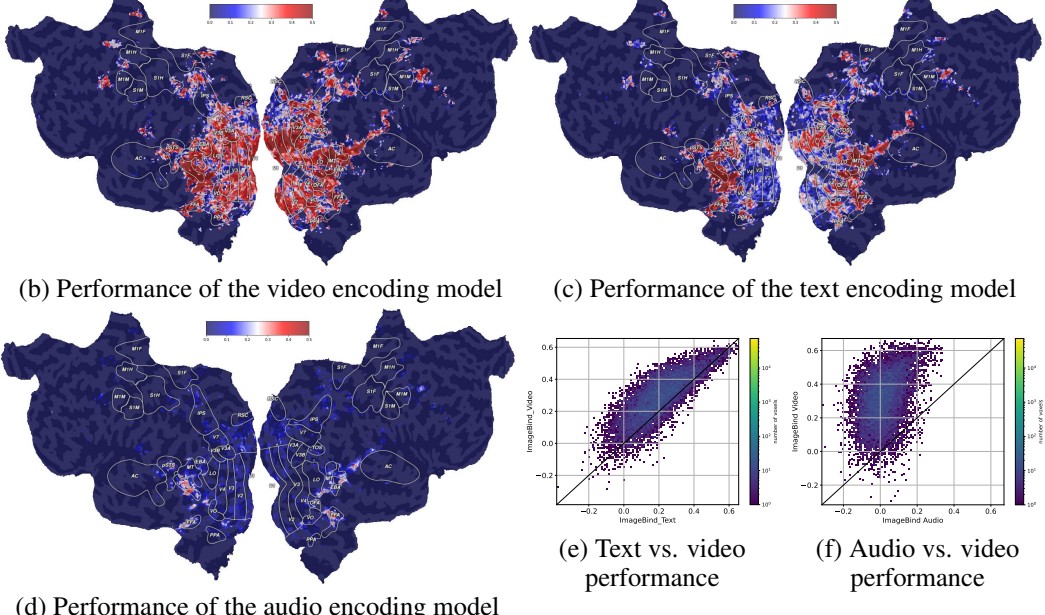

(b) Performance of the video encoding model

(c) Performance of the text encoding model

(d) Performance of the audio encoding model

(e) Text vs. video performance

(f) Audio vs. video performance

Figure 3: Prediction performance for each of the three encoding models for subject 1, projected on the subject's flattened cortical surface. Separate encoding models were estimated on brain responses to predict fMRI responses. Each modality performance is measured using Pearson correlation between predicted and actual responses.

mainly involved in multimodal tasks like various aspects of scene processing, and scene-category and object representations. In line with this study, we observe in Fig. 3(a) that for these regions both ImageBind Text and Video models are equally dominant (indicated by white color) compared to primary or early visual areas (which are mainly blue in color indicating that ImageBind Video model works better for those areas).

From Fig. 3(b), we observe that video representations predict all the regions very well (good spread of red color) compared to text (Fig. 3(c)) and audio (Fig. 3(d)) representations. The higher fMRI brain predictivity using text features (Fig. 3(c)) in high-level visual regions suggests that these voxels are significantly predicted and process semantic scene descriptions although the subject was only engaged in passive visual task. Further, the higher activity in MT region for ImageBind audio

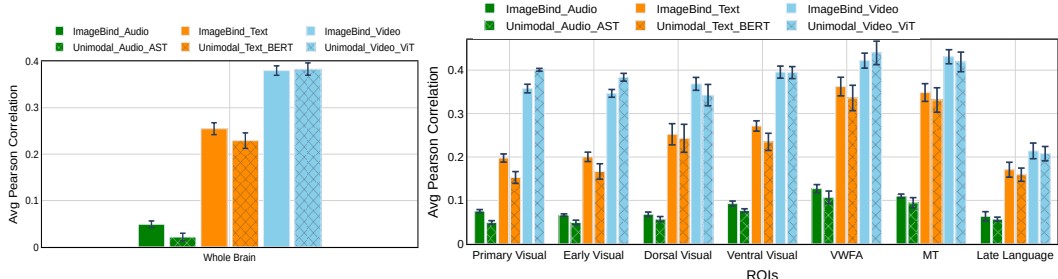

Figure 4: Brain alignment of separately estimated audio, language and vision encoding models averaged across all the subjects using ImageBind representations as well as Unimodal representations. The *left plot* compares the average Pearson correlation across all subjects and all voxels (i.e. whole brain). The *right plot* compares the region-wise performance of these models. Error bars indicate the standard error of the mean across participants.

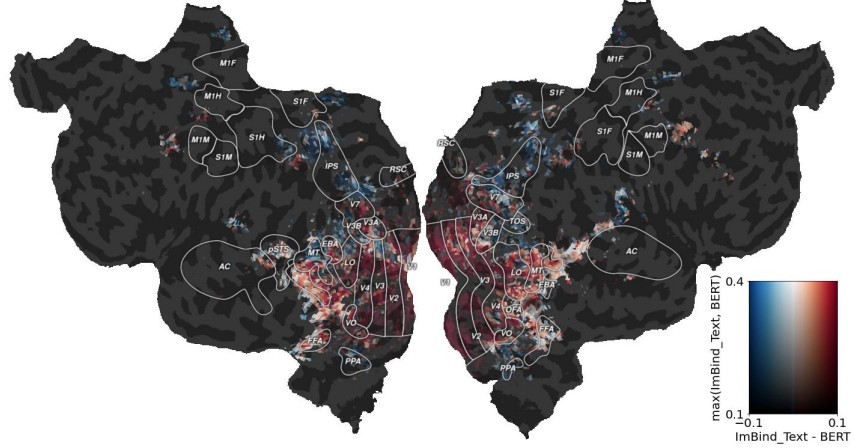

Figure 5: ImageBind text modality and Unimodal text prediction performance for each voxel in one subject as projected on the subject's flattened cortical surface. Voxels appear red if they are better predicted by ImageBind text, blue if they are better predicted by BERT features, and white if they are well predicted by both.

features (Fig. 3(d)) denotes that MT/MST responds to sound, recent evidence also suggests that the auditory response of MT/MST is selective for motion (Bedny et al., 2010).

From Fig. 3(e) and (f), we observe that for most voxels, video representations are better than text and audio. However, there are still 16% and 3% voxels where the text and audio representations are better than video respectively.

## 4.2 MULTIMODAL VS. UNIMODAL REPRESENTATIONS

To compare the multimodal performance with unimodal representations, we estimated language encoding models using BERT (Devlin et al., 2019), vision encoding models using ViT (Dosovitskiy et al., 2020) and audio encoding models using AST (Baade et al., 2022). Since these unimodal transformers are also used for feature extraction in ImageBind, they provide a good baseline to compare how well audio, language and visual features are aligned prior to and after binding to the joint embedding space. We use these unimodal features to evaluate their Pearson correlation for the fMRI response prediction. For the joint multimodal models and unimodal models, we report the predicted fMRI brain activity averaged across all the subjects at region-level in Fig. 4.

From Fig. 4, we make the following observations. (i) Binding helps both audio and text model predictions to perform significantly better than unimodal text and audio representations across different brain regions. Specifically, the joint multimodal model (ImageBind) text representations have an improved brain alignment in early visual, dorsal, VWFA and MT regions. This observation suggests

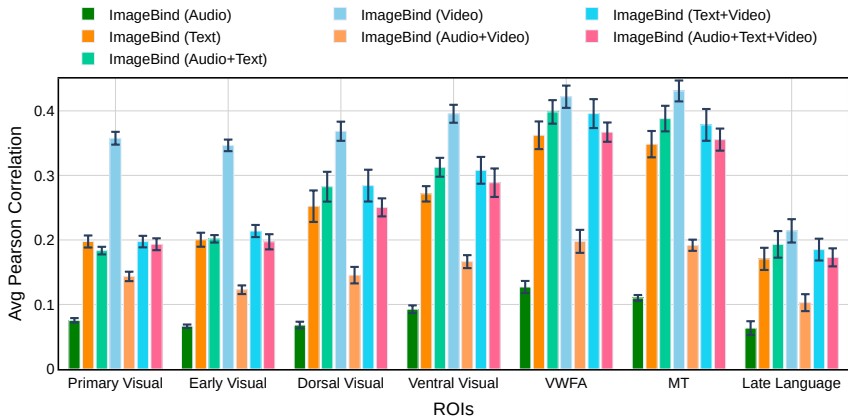

Figure 6: Brain alignment of separately estimated audio, text and video encoding models using ImageBind representations. We also show brain alignment of encoding models trained on additive combinations of the 3 ImageBind representations. The plot compares region-wise performance of these models averaged across all the subjects. Error bars indicate the standard error of the mean across participants.

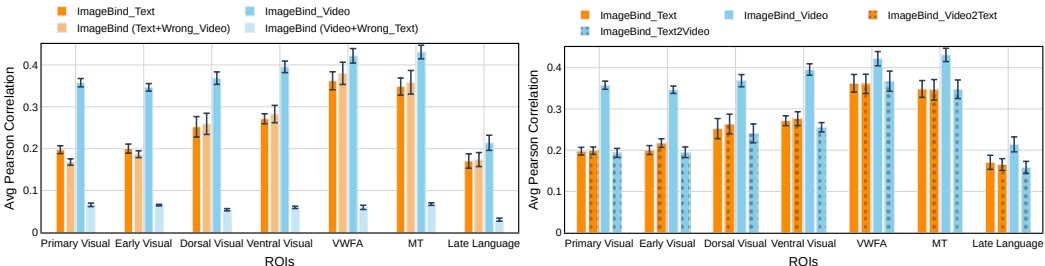

Figure 7: Brain alignment of separately estimated language and vision encoding models averaged across all the subjects using ImageBind representations as well as perturbed representations. The *left plot* compares the average Pearson correlation across all subjects and for each region. Error bars indicate the standard error of the mean across participants. The *right plot* compares the region-wise performance of the cross-model predictions.

that the integration of multiple modalities leads to the transfer of information from one modality to another, resulting in improved brain predictability. Based on the these, it can be inferred that these multimodal ImageBind-based models are indeed capable of learning multimodal connections that are relevant to the brain.

On the other hand, the joint multimodal model (ImageBind) video representations have higher brain predictivity only in the dorsal and MT regions than Unimodal video representations. This is a clear indication that the dorsal pathways are mainly active for object relations and actions (i.e. semantic text modality is helping the ImageBind video model to encode better fMRI predictivity for dorsal pathways and vice versa). Further, the higher predictivity obtained for the ImageBind text and ImageBind video models in the MT region corroborates its functionality, i.e. MT region is modulated by concordant auditory, text and visual input, albeit the passive nature of the visual stimuli, which argues for a role of this region in multimodal motion integration, beyond the pure visual domain (Çelik et al., 2021; Bedny et al., 2010; Mineault et al., 2021).

To compare the performance of ImageBind text model and Unimodal text model at a voxel-level for subject 1, we visualize the performance differences by projecting onto the subject's flattened cortical surface, as shown in Fig. 5 (see Appendix for flatmaps for other subjects). We observe that regions such as V3A, V3B, pSTS, FFA, and MT regions have higher brain scores for ImageBind (multimodal) text than BERT text features, which indicates that binding with other modalities yields an improved brain predictivity score. However, higher language regions display similar brain predictivity for both multimodal and unimodal text features suggesting that late language region voxels have no effect on integration with other modalities.

### 4.3 ADDITIONAL ANALYSIS

**Additive Combination of ImageBind embeddings**

Each of the three (text, audio and video) ImageBind representations already capture influence from other modalities. As we have observed so far, this has led to strong text and video models for the fMRI response prediction even when the stimuli was video. Next, we want to train an encoding model which uses a combination of per-modality ImageBind representations. In Fig. 6, we show region-wise comparison of brain alignment using individual ImageBind models versus encoding models trained on additive combinations of the three ImageBind representations. From the figure, we can see that the augmentation of the ImageBind Text to the ImageBind Video representations outperforms the ImageBind Text alone in all of the ROIs. This result is obvious because the stimuli to the subjects were video clips. Similarly, we can see that the augmentation of ImageBind Audio to the ImageBind Text representations, outperforms ImageBind Text alone, primarily, in the Dorsal Visual, Ventral Visual, VWFA and MT regions. Combination of any other modality to ImageBind Audio representations improves its performance; while combination to ImageBind Video representations reduces its performance, as expected. Combination of all modalities leads to a model better than other combinations except for using video alone. Overall, combining other modalities leads to improvements for text and audio but not for video.

**Perturbation Analysis** What happens if we feed the model the correct video but randomize the text captions in the input, and train the encoding model to predict the correct video-elicited brain recording? Also, what happens if we do vice versa, i.e., randomize the video but provide the correct text caption as input? Fig. 7(left) shows the results across different regions. We observe that in both cases, such perturbations lead to reduced correlation values. However, the reduction is much more when we use wrong text caption compared to the case when we use wrong video. From this we can conclude that the ImageBind text representations embed rich semantic scene descriptions although the subjects are involved in passive watching.

**Cross-Modal Performance** ImageBind text, audio and video representations are three different representations of the same underlying concept. Are they replaceable? To verify this, we train an encoding model using text but give it the video representation at test time. We refer to this model as Text2Video. Similarly, we also experiment with video at train and text at test time. We refer to this model as Video2Text. From Fig. 7(right), we observe that the best results are obtained only when we train and test both on the video representation. Training on text and testing on video or training on video and testing on text does not help. That said, we observe that model trained on video is comparable to model trained on text, when evaluated using text at test time. This leads us to infer that video representations subsume the semantic information present in text representations.

## 5 DISCUSSION

Full understanding of video stimuli requires humans to naturally integrate information from different senses (or modalities). Hence, brain encoding models should necessarily incorporate such multimodal fusion mechanisms. Thus, in this work, we leverage ImageBind to learn a unified representation space and bind semantics across video clips, text captions, and synthesized audio. We believe that the joint embedding space could lead to better results because it helps leverage cross-modality complementary information effectively, reduces modality-specific noise, and facilitates knowledge transfer between modalities. Empirically, we observe that brain encoding models trained using such ImageBind representations do lead to improved brain alignment in regions where multimodal integration happens. While the joint representations corresponding to text and video modalities exhibited better alignment, those of audio did not show significant performance in the early auditory areas (see the Appendix Fig. 13). This is understandable as the original video stimulation did not have audio. Further, it appears that any inner speech evoked due to video presentation might not have influenced early speech perception areas but aligned more with late language areas. However, these findings require careful future exploration by gathering multimodal fMRI datasets that emphasize interaction among modalities rather than passive viewing, to better understand multimodal processing in the brain. Future experiments could also incorporate more recent video captioning, audio synthesis and multimodal representation models. Overall, the experimental findings and computational results reported here indicate that the visual system in the brain may be converting visual input implicitly into comprehensive semantic scene descriptions.

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

# A   IMAGEBIND TEXT VS VIDEO FOR OTHER SUBJECTS

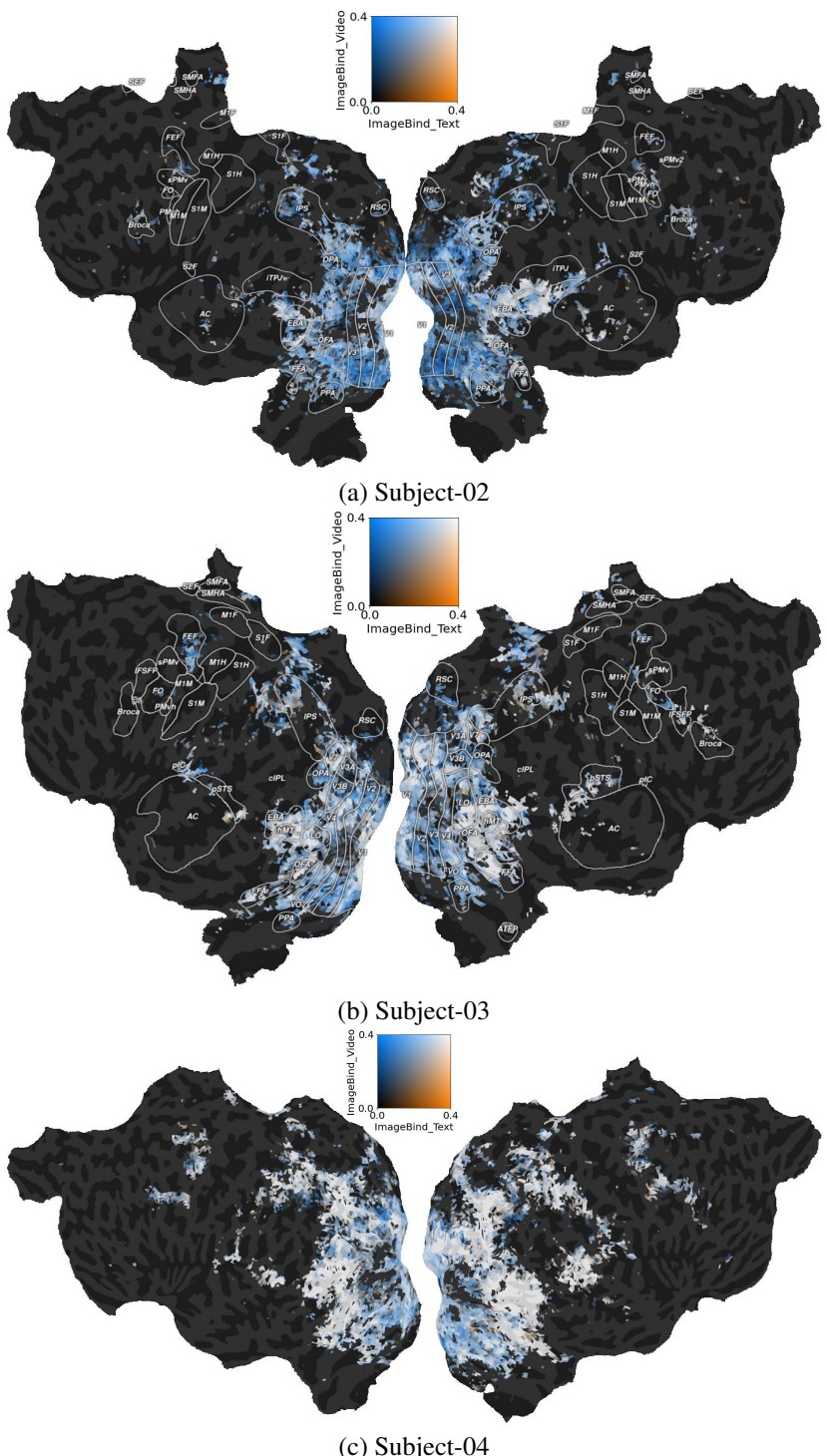

(a) Subject-02

(b) Subject-03

(c) Subject-04

Figure 8: Contrast estimation between ImageBind text and video performance for each voxel for remaining participants (subjects 02, 03 and 04). Voxels appear orange if they are better predicted by text, blue if they are better predicted by video features, and white if they are well predicted by both.

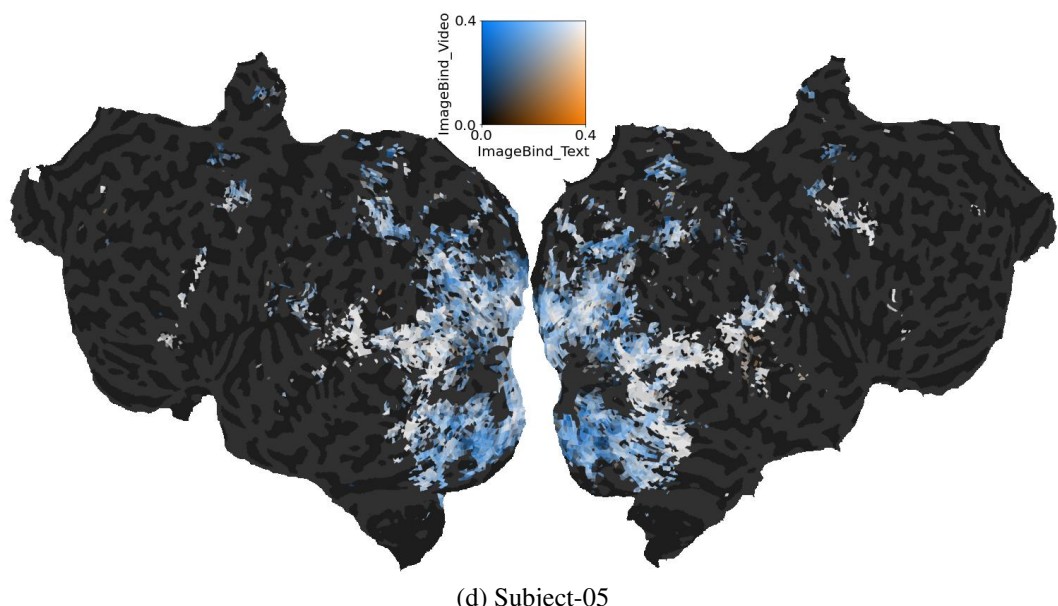

(d) Subject-05

Figure 9: Contrast estimation between ImageBind text performance and video performance for each voxel for subject 05. Voxels appear orange if they are better predicted by text, blue if they are better predicted by video features, and white if they are well predicted by both.

## B    MULTIMODAL VS. UNIMODAL FLATMAPS

### B.1    IMAGEBIND VIDEO VS. UNIMODAL VIDEO VITH

## C    MULTIMODAL BRAIN PROCESSING REGIONS

The following brain regions have been known to participate in multimodal processing.

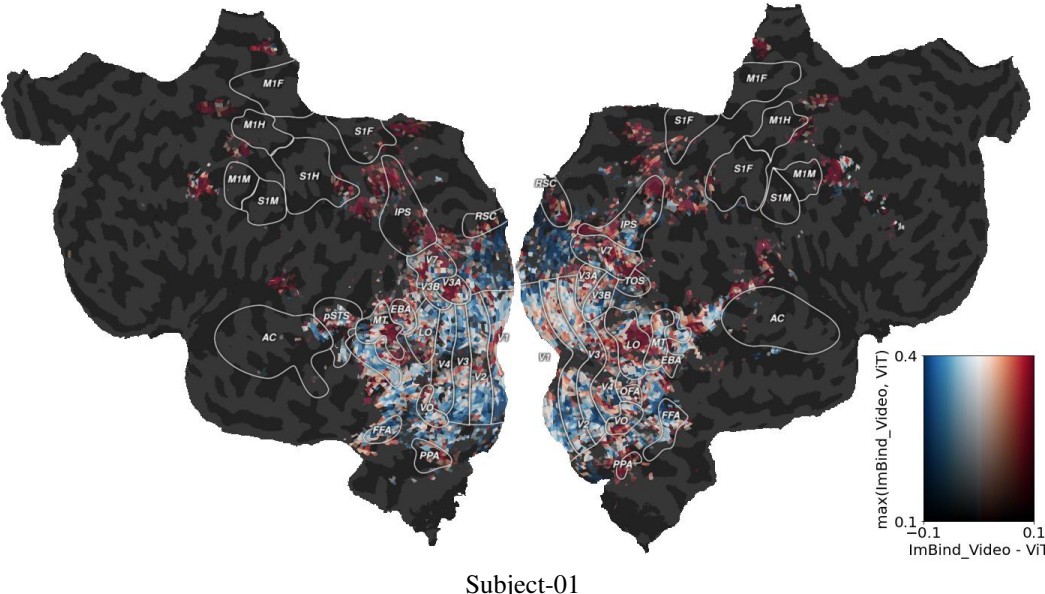

Subject-01

Figure 10: ImageBind video modality and Unimodal video prediction performance for each voxel in corresponding subject as projected on the subject's (subject-01) flattened cortical surface. Voxels appear red if they are better predicted by ImageBind text, blue if they are better predicted by BERT features, and white if they are well predicted by both.

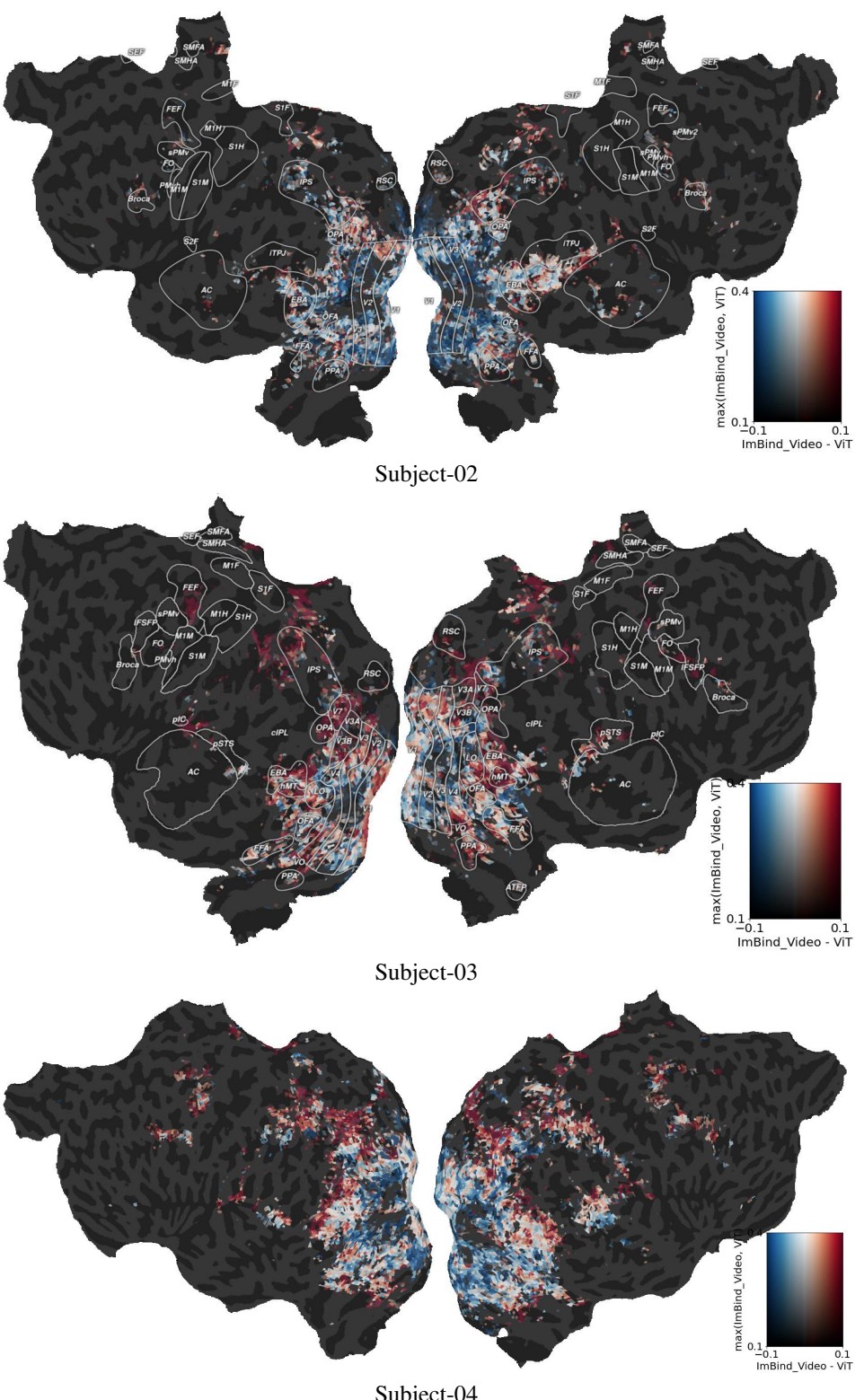

Subject-02

Subject-03

Subject-04

Figure 11: ImageBind video modality and Unimodal video prediction performance for each voxel in corresponding subject as projected on the subject's flattened cortical surface. Voxels appear red if they are better predicted by ImageBind text, blue if they are better predicted by BERT features, and white if they are well predicted by both. Results for subjects 02, 03 and 04.

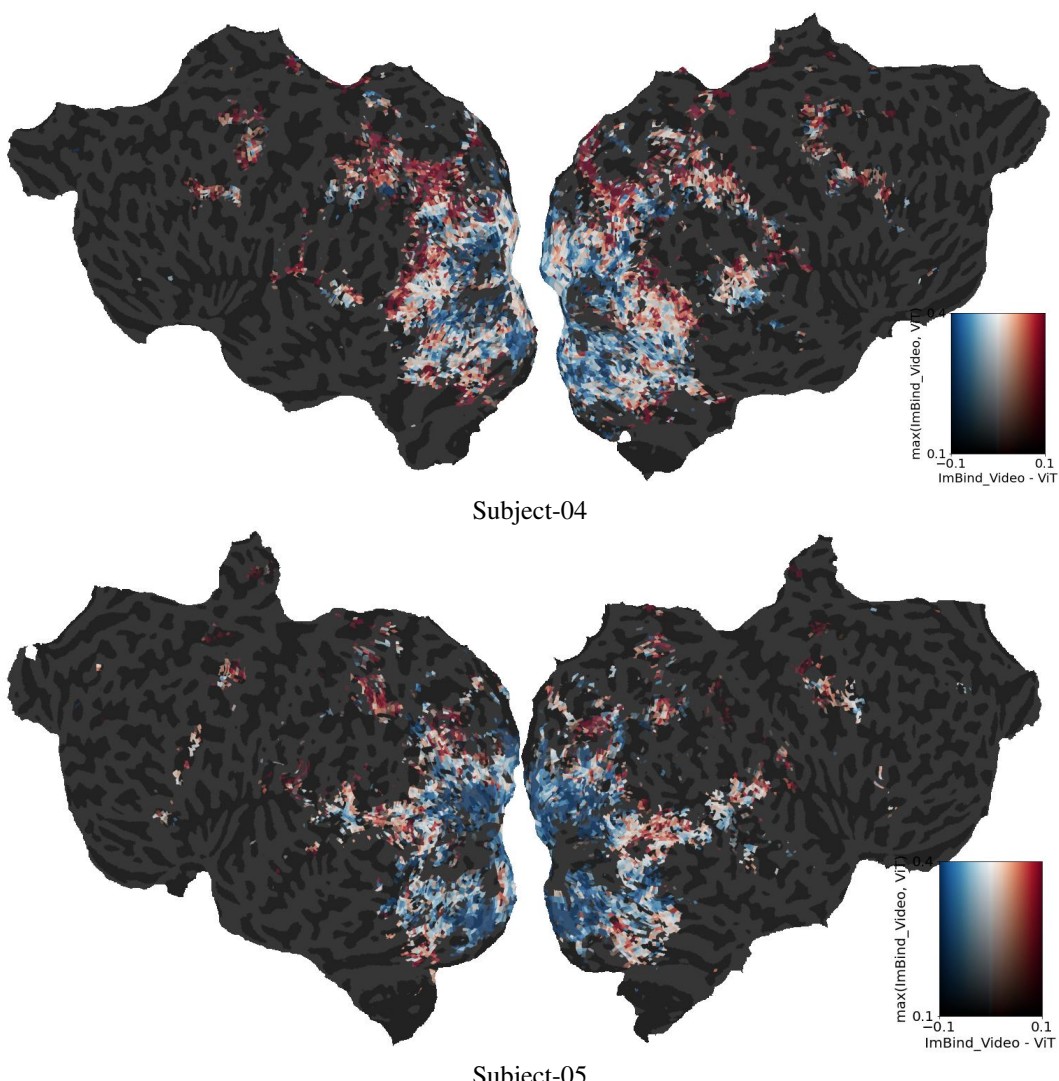

Subject-04

Subject-05

Figure 12: ImageBind video modality and Unimodal video prediction performance for each voxel in corresponding subject as projected on the subject's flattened cortical surface. Voxels appear red if they are better predicted by ImageBind text, blue if they are better predicted by BERT features, and white if they are well predicted by both. Results for subjects 04 and 05.

- Superior Temporal Sulcus (STS): Involved in integrating visual and auditory information, especially related to social cues.
- Posterior Parietal Cortex (PPC): Plays a role in integrating visual, tactile, and proprioceptive information for spatial perception and coordination.
- Inferior Parietal Lobule (IPL): Involved in integrating sensory information for actions and object recognition.
- Ventral Intraparietal Area (VIP): Integrates visual and somatosensory information for spatial awareness.
- Prefrontal Cortex: The prefrontal cortex, particularly the dorsolateral prefrontal cortex (DLPFC) and the ventromedial prefrontal cortex (vmPFC), is involved in higher-order cognitive processes that require integrating information from various modalities for decision-making, problem-solving, and planning.
- Hippocampus: The hippocampus is important for forming spatial and episodic memories by integrating information from different sensory modalities into a coherent memory.

| Region | Functional Description |
|---|---|
| PV: V1 | The primary visual cortex is the earliest cortical region for visual processing. It processes basic visual features, such as edges, orientations, and spatial frequencies. Lesions in V1 can lead to blindness in the corresponding visual field. |
| EV: V2, V3, V4 | These early visual areas are involved in further processing of visual information. V2 processes more complex patterns than V1. V3 is involved in processing motion and form. V4 is vital for color recognition and processing complex shapes. |
| DV: V3A, V3B, V6, V6A, V7, IPS1 | Dorsal visual areas focus on the "where" pathway of vision, processing spatial location, motion, and guiding motor actions. V3A and V3B are associated with motion processing, while V6 and V6A integrate visual and somatic inputs to inform motor actions. V7 and IPS1 are involved in visual attention and mapping objects in space. |
| VV: V8, VMV1, VMV2, VMV3, VVC, FFC | Ventral visual regions are part of the "what" pathway, focusing on object and face recognition. V8 has roles in color perception. The VMV clusters are involved in advanced visual processing and recognition, and the FFC is linked with recognizing faces and detailed object recognition. |
| VWFA: PH, TE2P | The visual word form area specializes in recognizing written words and letters, facilitating the transition from visual representations of words to their associated meanings and sounds. This region is crucial for skilled reading. |
| MT: Middle Temporal | The middle temporal regions specialize in processing visual motion. MT and MST play roles in detecting and analyzing moving objects in our environment. The LO clusters are associated with object recognition, especially in terms of shapes. FST aids in the integration of motion and form, while V3CD contributes to depth perception. |
| Late Language | Late language regions contribute to various linguistic processes. Areas 44 and 45 (Broca's area) are vital for speech production and grammar comprehension. The IFJ, PG, and TPOJ clusters are involved in semantic processing, syntactic interpretation, and discourse comprehension. STGa and STS play roles in phonological processing and auditory-linguistic integration. TA2 is implicated in auditory processing, especially in the context of language. |

Table 2: Detailed functional description of various brain regions.

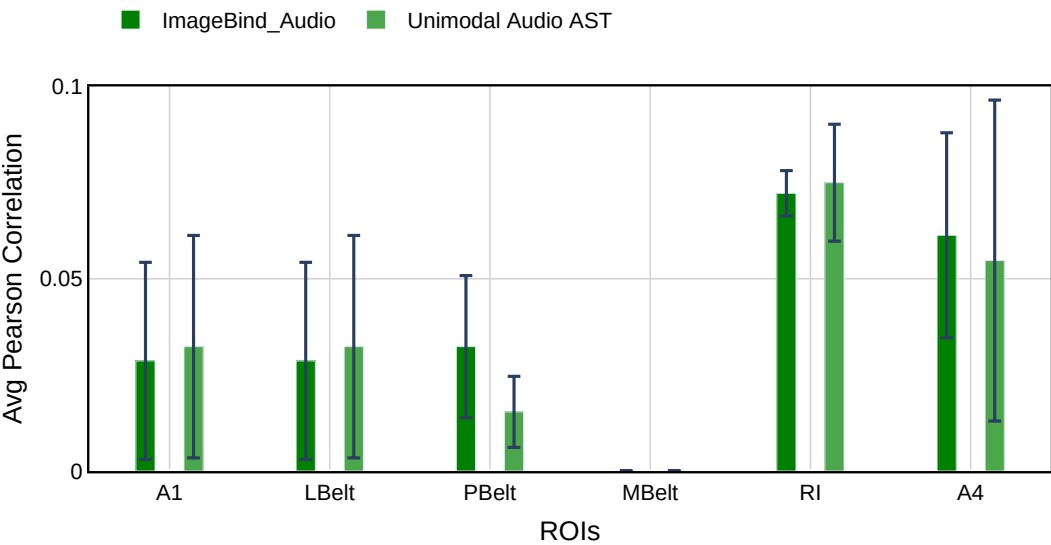

Figure 13: Brain alignment of separately estimated ImageBind audio and unimodal audio encoding models averaged across all the subjects. Error bars indicate the standard error of the mean across participants.

