# OpenReview forum: "Brain encoding models based on binding multiple modalities across audio, language, and vision"
_ICLR.cc/2024/Conference — Submitted to ICLR 2024_

### Official Review · Reviewer_77Kr · 2023-10-29

**Soundness:** 1 poor
**Presentation:** 1 poor
**Contribution:** 1 poor
**Rating:** 1
**Confidence:** 5

**Summary:**

The authors examine whether activations extracted from multimodal neural networks can fit signals derived from functional magnetic resonance imaging (fMRI) measurements obtained while participants watch videos. The paper also explores algorithms that generate captions, convert captions to speech and then use text, audio and video information to try to fit fMRI responses. Mysteriously, the authors conclude that the role of the visual system is to extract semantic informaiton.

**Strengths:**

The question of studying the similarities and differences between different modalities is important and worth studying.

It would be very useful for the field to understand how the brain represents visual, auditory, and text information (but this is not studied in the current paper).

**Weaknesses:**

Before jumping onto showing correlations (Fig. 2), it would be useful to show the actual activations both in the networks and in the fMRI signals to better understand how the correlations are computed.

The basic norms of scientific reporting are not followed here. Axis should be labeled, error bars should be defined.

It would also be useful to spell out the number of features and training used in each case. Does the order in Fig. 2 reflect the number of features or the amount of training in each modality or the successes of the neural network models in each modality?

The features are correlated and therefore it is hard to deduce anything from the fitting analyses. For example, if there is a ball in the image, and the text says ball and the audio says ball, then one can find that language areas can be fit by "visual" features but this does not mean that the language areas represent visual features. Conversely, visual areas can be fit by text, not because visual areas represent text. To understand the relationship between different modalities, we need rigorous controlled experiments that can prove uncorrelated feature dimensions. Unfortunately, this problem is ubiquitous throughout the paper.

There are no comparisons with different baselines, different neural network models, ablation studies.

**Questions:**

Are the videos shown with sound? If so, why not use the actual sound and caption from the video?

What is the point of converting caption to speech? In the best case scenario, the caption to speech is perfect and the information is redundant. In the worst case scenario, the speech is a bad rendering of the caption and merely adds noise.

It would be useful to conduct experiments where there is only visual information that is dissociated from audio information and from language information, experiments with only language information, etc. Even better, one could run experiments where different modalities are orthogonalized (e.g. show a ball and present the word chair). Once the modalities are rigorously decorrelated, it may be possible to begin to disentangle the contribution of different modalities to brain signals.

---

> ### Author Response · Authors · 2023-11-17
>
> **Overall contributions of our paper**
>
> Before we respond to individual queries, we thought of summarizing the contributions and novelty of the paper to give a broader context. Understanding how neural network representations are related to human brain representations has been a central question in deep learning and our paper makes an important addition to the literature in this aspect. It is true that prior works already used similar frameworks (using representations from transformers to build encoders that predict fMRI data/images). However, our work makes significant novel contributions in demonstrating how to leverage joint information across modalities for better brain alignment as mentioned in brief in the following:
>
> * While earlier works investigated alignment with image+captions and image+language features from language models (LM), for the first time, we investigated how the joint representations learned from video+captions+audio align with the brain responses.
> * We use the recent popular ImageBind model to learn a unified representation space that binds semantics across video clips, text captions, and synthesized audio.
> * Our results verify our hypothesis that the joint embedding space leads to better alignment as compared to alignment with unimodal representations. This is possible due to leveraging cross-modality complementary information effectively, reducing modality-specific noise, and facilitating knowledge transfer between modalities.
> * It is to be noted that the movie clips were presented silently to the subjects and there is no audio or subtitles (captions) available as the movies were constructed by concatenating sequences of 10-20s video clips.
> * There are unfortunately no high-quality captions readily available for the short-clips dataset [1]. Thus, the captions and the synthesized audio for the captions form an additional data contribution to the video clips dataset.
>
> [1] Tang, J., Du, Meng., & Huth, A. G. Brain encoding models based on multimodal transformers can transfer across language and vision, NeurIPS (2023).
>
> **Before jumping onto showing correlations (Fig. 2), it would be useful to show the actual activations both in the networks and in the fMRI signals to better understand how the correlations are computed.**
>
> We understand the reviewer's standpoint about investigating how the brain responds to video clips and how the deep networks process video information before looking at the performance with joint representations. Although these basic questions are very important, they have already been addressed in previous works [1], [2], [3]. They have become important standard results in computational cognitive neuroscience. Consequently, we focused on presenting the correlation results indicating the accuracy of joint representations, as shown in Fig 2, along with brain surface visualizations in Fig 3. Further, we elaborate on the associated methods relevant to our computations that differ from previous studies in the methods section.
>
> [1] Nishimoto, S. et al. Reconstructing visual experiences from brain activity evoked by natural movies. Curr. Biol. 21, 1641–1646 (2011).
>
> [2] Huth, A. G. et al. Decoding the Semantic Content of Natural Movies from Human Brain Activity. Front. Syst. Neurosci. 10, 81 (2016).
>
> [3] Popham, S. F. et al. Visual and linguistic semantic representations are aligned at the border of the human visual cortex. Nat. Neurosci. 24, 1628–1636 (2021).
>
> **The basic norms of scientific reporting are not followed here. Axis should be labeled, error bars should be defined.**
>
> Thanks for your concern. We believe that we have followed the scientific norms for all the figures in our paper. We have clearly defined the plot labels and the axes. Also, we have mentioned the implications of the error bars wherever necessary.
> * For instance, in Figure 2 main paper, the x-axis represents the whole brain and the y-axis represents the Average Pearson Correlation values. The different colors (legend)  correspond to the ImageBind Video (blue), ImageBind Text (orange), and ImageBind Audio (green).
> * Similarly, for Figure 2 (right), the x-axis represents the ROIs (Primary Visual, Early Visual, Dorsal Visual, Ventral Visual, VWFA, MT and Late Language), and the y-axis represents the Average Pearson Correlation values. The different colors (legend)  correspond to the ImageBind Video (blue), ImageBind Text (orange), and ImageBind Audio (green). All of these are already defined in the paper. The same holds for other figures. It would be great if you could clearly identify which axis label is missing and in which figure. We would be happy to clarify further.

---

> > ### Author Response · Authors · 2023-11-17
> >
> > **Are the videos shown with sound? If so, why not use the actual sound and caption from the video?**
> >
> > Thank you for this question. It appears that there is some confusion about the nature of the stimuli that the participants have experienced during fMRI acquisition. We reiterate that the movie clips were presented silently to the subjects and there was no audio or subtitles (captions) available with the clips. We have already mentioned this in the submitted draft, specifically in the discussion section. However, to avoid any potential confusion, we shall add this in the dataset curation section as well and make it clearer in the camera-ready version (Please see the preamble part of the response).
> >
> > **What is the point of converting caption to speech? In the best case scenario, the caption to speech is perfect and the information is redundant. In the worst case scenario, the speech is a bad rendering of the caption and merely adds noise.**
> >
> > * We agree with the reviewer’s general point about possible redundancy between captions and synthesized audio. Just as a reminder, we would like to emphasize that the movie clips were presented silently to the subjects and no audio or subtitles (captions) were present.
> > * In the current formulation, we learn joint representations associated with audio+text+video from ImageBind. We compare the brain alignment with unimodal audio representations against that with the joint representations.
> > * We observe that the aligned representations (using the ImageBind model)  have improved brain alignment in multimodal semantic processing regions in the brain (such as Dorsal Visual, MT, and Late Language regions).
> > * Although audio representations yield low correlations (Fig. 2), from Figure 13 (Appendix) it appears that any inner speech evoked due to video presentation might not have influenced early speech perception areas but aligned more with late language areas.
> > * Further, from Figure 6, we can see that the augmentation of ImageBind-Audio to ImageBind-Text representations outperforms ImageBind-Text alone, primarily in the Dorsal Visual, Ventral Visual, VWFA and MT regions.
> > * The combination of any other modality to ImageBind-Audio representations improves its performance.
> > * All these are interesting additional results and are made possible only because of the availability of features learned from the synthesized audio and cross-comparison analysis with those from other modalities.
> >
> > **It would also be useful to spell out the number of features and training used in each case. Does the order in Fig. 2 reflect the number of features or the amount of training in each modality or the successes of the neural network models in each modality?**
> >
> > * Thank you for the question. We want to clarify that the number of features for each modality is the same as provided by the ImageBind model.
> > * We extract common embedding representations for the brain encoding task. We input video, synthesized audio and the generated text caption at each TR (repetition time) and obtain the aligned embeddings for the three modalities.
> > * We use the pre-trained ImageBind Transformer model which outputs a 1024 dimensional representation for each of the three modalities. Hence, the number of features and training methodology is the same for the three modalities.
> > * There are no manually curated features. Therefore, the results reported in Fig. 2 do not depend on the number of features and the amount of training in each modality, but depend on the alignment of modality representations with the brain response.

---

> > > ### Author Response · Authors · 2023-11-17
> > >
> > > **The features are correlated and therefore it is hard to deduce anything from the fitting analyses. For example, if there is a ball in the image, and the text says ball and the audio says ball, then one can find that language areas can be fit by "visual" features but this does not mean that the language areas represent visual features. Conversely, visual areas can be fit by text, not because visual areas represent text. To understand the relationship between different modalities, we need rigorous controlled experiments that can prove uncorrelated feature dimensions. Unfortunately, this problem is ubiquitous throughout the paper.**
> > >
> > > * We agree with the reviewer’s basic logic and the example, however what is common across different modalities representing “ball” is the concept of “ball” and the associated semantic information.
> > > * We claim that this is precisely what is learned by the joint embedding space obtained from ImageBind that binds the semantic information across video, text, and audio modalities. If indeed the features are all correlated we should see similar brain alignment performance across different modalities.
> > > * We can clearly see that this is not the case (Fig. 2), indicating that the joint representations captured by different modalities are not correlated but are aligned in the common representational space. CLIP [1] and other efforts [2] have started looking at the power of joint representations.
> > > * A very recent study [3] has gone further and utilized language encoding models to predict responses to videos and image encoding models to predict brain responses for listening to stories in a cross-modal prediction task [1].
> > > * Motivated from this work, we use ImageBind as a useful tool to bridge across modalities and to aid our understanding of the brain with visual stimuli.
> > > * Given that the brain representations are associative across modalities, the current effort advances the state-of-the-art in investigating the usefulness of cross-modal representations in explaining the brain activation, especially in the multimodal association regions of the brain.
> > >
> > > [1] Wang, A. Y., Kay, K., Naselaris, T., Tarr, M. J. & Wehbe, L. Incorporating natural language into vision models improves prediction and understanding of higher visual cortex. bioRxiv 2022.09.27.508760 (2022) doi:10.1101/2022.09.27.508760.
> > > [2] Oota et al. 2022, Visio-Linguistic Brain Encoding, COLING 2022
> > > [3] Tang, J., Du, Meng., & Huth, A. G. Brain encoding models based on multimodal transformers can transfer across language and vision, NeurIPS (2023).
> > >
> > > **There are no comparisons with different baselines, different neural network models, ablation studies.**
> > >
> > > In terms of different baselines:
> > > * We compare ImageBind-Text, ImageBind-Audio and ImageBind-Video with Unimodal_Text_BERT, Unimodal_Audio_AST, and Unimodal_Video_ViT models, respectively for the whole brain and also for various brain regions of interest (ROIs).
> > > * Unfortunately since there is no other previous work in this area, a strong baseline is not currently available for us to compare with.
> > >
> > > In terms of ablation studies:
> > > * We have conducted perturbation analysis where we feed the correct video to the model but randomize the text captions in the input, and train the encoding model to predict the correct video-elicited brain recording.
> > > * Also, we investigate what happens if we do vice versa, i.e., randomize the video but provide the correct text caption as input. Fig. 7 (left) shows the results across different regions.
> > > * We observe that in both cases, such perturbations lead to reduced correlation values. However, the reduction is much more when we use the wrong text caption compared to the case when we use the wrong video.
> > > * From this, we can conclude that the ImageBind-Text representations embed rich semantic scene descriptions although the subjects are involved in passive watching.
> > >
> > > We have conducted cross-modal retrieval analysis.
> > > * ImageBind text, audio, and video representations are three different representations of the same underlying concept. Are they replaceable?
> > > * To verify this, we train an encoding model using text but give it the video representation at test time. We refer to this model as Text2Video.
> > > * Similarly, we also experiment with video at train and text at test time. We refer to this model as Video2Text. From Fig. 7 (right) in the manuscript, we observe that the best results are obtained only when we train and test both on the video representation.
> > >  * Training on text and testing on video, or training on video and testing on text does not help. That said, we observe that a model trained on video is comparable to a model trained on text, when evaluated using text at test time. This leads us to infer that video representations subsume the semantic information present in text representations.

---

### Official Review · Reviewer_bZKi · 2023-10-31

**Soundness:** 3 good
**Presentation:** 3 good
**Contribution:** 3 good
**Rating:** 8
**Confidence:** 5

**Summary:**

The paper evaluates the effectiveness of multimodally aligned features in understanding fMRI brain responses to videos. Using a dataset of fMRI scans from subjects watching videos, the study generates multi-event captions and synthesized audio to create a joint embedding across audio, text, and video. This joint embedding trains models to predict fMRI responses. Key findings indicate that the visual system primarily focuses on converting visual input into semantic descriptions, and multimodal alignment enhances the prediction of brain activity compared to unimodal approaches.

**Strengths:**

- The authors provides concise interpretability of their model's performance. They relate the outputted embeddings to specific regions of the brain, giving good intuition of how the human responds cognitively to external stimuli.
- The authors provide sufficient ablation study to identify each modalities affect on performance.

**Weaknesses:**

Maybe the authors can use another metric (MSE) to quantify the error in the fMRI activity prediction.

**Questions:**

For future work, have the authors considered using different models for the text, video, and audio encoder to validate whether these findings generalize across different models as well?
It would also be interesting work to do a canonical correlation analysis to measure the relationship between the generated joint embeddings and the fMRI signals.

---

### Official Review · Reviewer_TnaU · 2023-11-01

**Soundness:** 2 fair
**Presentation:** 3 good
**Contribution:** 2 fair
**Rating:** 3
**Confidence:** 5

**Summary:**

This paper uses multimodally aligned features from visual, auditory and semantic domain to build encoding models to predict fMRI response to silent videos. The paper compares encoding model performances across different single modality models and multimodal models and showed that alignment help with brain prediction.

**Strengths:**

The paper is presents an interesting idea of leveraging multimodal alignment to probe the multimodal representation in the human brain. The idea itself is relatively novel.

The paper is written clearly.

**Weaknesses:**

The findings of the paper lack novelty. The main message from the results, namely, visual perception carries semantic information that can be predicted with semantic-based encoding model, is well known and demonstrated across modalities and with different models.

Given the above, it is not surprising to me that other modality that are not present in the stimuli could individually predict brain responses to some extend (and that they each perform worse than the visual model), since the extracted feature from these modalities all share the semantic information contained in the visual feature and in the brain. It is also not surprising that, with the ImageBind model each modality gains a small boost in performance because of more semantic information they gain from the visual modality.

Since the authors didn’t make comparison between imageBind models to other single or multi- modal models, it is unclear whether ImageBind help at all in terms of overall brain prediction.

Throughout the paper, comparison of models are largely limited to univariate bar plots or contrasts plots in brain maps. Without analysis like variance partitioning or feature regression between model space it is unclear whether these models from different modalities indeed predicts any regions uniquely.

ImageBind as a useful tool to bridge across modalities, perhaps has more potentials in aiding our understanding of the brain with stimuli that actually consists of multiple modalities (movies with actual audios). With those one could study, for example, difference of subtitles vs video transcripts encoding in the brain with presence of visual input.

**Questions:**

In Figure 1, I think the colors of the arrow or the order of the modality are flipped.

---

### Author Response · Authors · 2023-11-21

Dear Reviewers,

We thank you for the insightful comments on our work. Your suggestions have now been incorporated in our revision and we are eagerly waiting for your feedback.

As the author-reviewer discussion phase is approaching its conclusion in just a few hours, we are reaching out to inquire if there are any remaining concerns or points that require clarification. Your feedback is crucial to ensure the completeness and quality of our work. Your support in this final phase, particularly if you find the revisions satisfactory, would be immensely appreciated.

Regards,
Authors

---

### Meta-Review · Area_Chair_kAp9 · 2023-12-14

**Metareview:**

This work aims to study the contributions of various stimuli to study mutli-modal representations in the brain. The overall approach is to construct a predictive model that can reconstruct fMRI data of subjects watching a silent video based on audio, video, and caption data. The authors show the capability of multi-modality based predictors to outperform unimodal models. There were split opinions from the reviewers. Strengths were noted as to the clarity of the writing and importance of the study. Weaknesses were noted in the data reporting, limited comparisons, and figure clarity. Unfortunately these weaknesses were not resolved during the discussion period and I am recommending the paper not be accepted.

**Justification For Why Not Higher Score:**

Specific weaknesses identified by the reviewers were:
 - data reporting, i.e., demonstrating the validity of the fMRI data via response characterization
 - limited comparisons to other methods
 - figure clarity in reporting results

**Justification For Why Not Lower Score:**

N/A

---

### Decision · Program_Chairs · 2024-01-16

Reject